# Role of Cine-Magnetic Resonance Imaging in the Assessment of Mediastinal Masses with Uncertain/Equivocal Findings from Pre-Operative Computed Tomography Scanning

**DOI:** 10.3390/diagnostics14151682

**Published:** 2024-08-02

**Authors:** Umberto Cariboni, Lorenzo Monti, Emanuele Voulaz, Efrem Civilini, Enrico Citterio, Costanza Lisi, Giuseppe Marulli

**Affiliations:** 1Division of Thoracic Surgery, IRCCS Humanitas Research Hospital, 20089 Milan, Italyemanuele.voulaz@humanitas.it (E.V.); 2Istituti Clinici Scientifici Maugeri IRCCS, 27100 Pavia, Italy; 3Department of Biomedical Sciences, Humanitas University, 20090 Milan, Italy; efrem.civilini@hunimed.eu (E.C.);; 4Division of Vascular Surgery, IRCCS Humanitas Research Hospital, 20089 Milan, Italy; 5Division of Cardiac Surgery, IRCCS Humanitas Research Hospital, 20089 Milan, Italy; 6Department of Diagnostic and Interventional Radiology, IRCCS Humanitas Research Hospital, 20089 Milan, Italy

**Keywords:** mediastinum, cine-magnetic resonance, malignant neoplasms, infiltration

## Abstract

Background: Malignant neoplasms originating from or involving the mediastinum represent a diagnostic and therapeutic challenge when they are in contact with nearby cardiovascular structures. We aimed to test the diagnostic accuracy of cine-magnetic resonance imaging (cine-MRI) in detecting the infiltration of cardiovascular structures in cases with uncertain or equivocal findings from contrast-enhanced Computed Tomography (CT) scanning. Methods: Fifty patients affected by tumors with a suspected invasion of mediastinal cardiovascular structures at the pre-operative chest CT scan stage underwent cine-MRI before surgery at our Institution. Intraoperative findings and the histological post-surgical report were used as a reference standard to define infiltration. Inter- and intra-observer agreement for CT scans and cine-MRI were also computed over a homogenous sample of 14 patients. Results: Cine-MRI had a higher negative predictive value (93% vs. 54%, *p* < 0.001) than CT scans, higher sensitivity (91% vs. 16%, *p* < 0.001), as well as greater accuracy (66% vs. 50%, *p* < 0.001) in detecting cardiovascular invasion. Cine-MRI also showed better inter- and intra-observer agreement for infiltration detection. Conclusions: Cine-MRI outperforms conventional contrast-enhanced chest CT scans in the preoperative assessment of cardiovascular infiltration by mediastinal or pulmonary tumors, making it a useful imaging modality in the preoperative staging and evaluation of patients with equivocal findings at the chest CT scan stage.

## 1. Introduction

Malignant tumors involving the mediastinum often represent a great diagnostic and therapeutic challenge for the surgeons because, in many cases, they get in contact with, either by compressing or invading, nearby cardiovascular structures such as the thoracic aorta, the pulmonary arteries and veins, the superior and inferior vena cava, the pericardium and the myocardium [1,2]. Surgery represents the mainstay of treatment in most primary lung and mediastinal tumors like non-small cell lung cancer (NSCLC) and thymic epithelial tumors (TET); however, correct diagnosis and staging, preoperative care and postoperative management have equal importance [3]. There are some reports of long-term survivors from extended resections of primary lung cancers involving the mediastinum (superior vena cava, aorta, left atrium and main pulmonary artery), but there are few data to support these extended operations [4]. However, it has been well established that selected patients with mediastinal invasion but without mediastinal node involvement can be viewed as candidates for surgery [5]. For advanced thymic epithelial tumors, a multidisciplinary strategy including radical surgery remains the widely accepted gold-standard treatment [6]. Various reports have demonstrated satisfactory outcomes for the surgical resection of thymic epithelial tumors (TET) when combined with the resection and reconstruction of major mediastinal vessels, particularly the superior vena cava and innominate veins. Nonetheless, the literature reveals a diverse array of surgical approaches and materials used for reconstruction [7]. 

An accurate preoperative radiological evaluation of the disease should aim to assess the resectability of the tumor even if the infiltration is present [8]. In the latter case, in fact, these neoplasms may still be amenable to surgical resection, provided that the appropriate equipment and multidisciplinary expertise are available [9]. The ability to make an accurate preoperative diagnosis of the infiltration of the mediastinal cardiovascular structures is fundamental to shaping the best surgical approach; extracorporeal circulation (ECC) can be used when the heart chambers are involved to allow aggressive surgical approaches, as well as dedicated surgeons (i.e., cardiovascular surgeons) and a properly equipped surgical theatre [10,11]. 

In these conditions, intraoperative airway management represents a crucial strategy for performing a radical and safe intervention. Extracorporeal membrane oxygenation (ECMO) is an effective ventilation strategy that provides cardiac or respiratory support without interference with the surgical field [12]. The ECMO circuit for an adult patient usually consists of an inflow cannula, a centrifugal pump, a heat exchanger embedded in a membrane oxygenator and an inflow cannula that transports arterialized blood [13,14]. Two different types of ECMO are available: veno-venous (VV)- and veno-arterial (VA)-ECMO. The use of ECMO in thoracic surgery is usually limited to certain procedures where adequate ventilation is not otherwise as feasible, such as surgery in patients with a history of previous extensive contralateral pulmonary resection, including pneumonectomy, and surgery in patients with an invasion of mediastinal structures [15]. Finally, ECMO is the extracorporeal life support of choice for the treatment of NSCLC presenting with carinal extension and requiring complex tracheo-bronchial reconstruction through an obstacle-free surgical field. The use of this technique in thoracic surgery allows hemodynamic stability with less risk for brain and myocardial oxygenation and a free operating field [16,17]. 

Contrast-enhanced CT scans and cine-MRI are imaging modalities used to define the real infiltration of mediastinal structures, with no existing guidelines favoring one over the other as the gold-standard method to evaluate infiltration [18]. Since 2008, at Humanitas Research Hospital, cine-MRI has been performed on all patients affected by primary mediastinal tumors or tumors involving the mediastinum who are potentially candidates for surgery, but have inconclusive CT scan findings, to refine pre-operative assessment and to define the best surgical approach. 

CT scans and cine-MRI potentially provide different and complementary information. CT scans are the first-line imaging modality in mediastinal mass evaluation, giving accurate information about the neoplasms’ characteristics and allowing for TNM staging of the tumor [19,20]. Despite being the gold-standard imaging modality for T assessment in neoplasms contacting nearby cardiovascular structures, contrast-enhanced chest CT scans show variable diagnostic performance in terms of sensitivity and specificity regarding infiltration detection, possibly producing some over- or under-staging [21,22]. Despite an inferior spatial resolution, chest cine-MRI provides useful high-contrast and temporal resolution images, possibly allowing for a better identification of tissue cleavage between the mass and nearby cardiovascular structures. Over the years, MRI technology has continued to evolve, and current cine-MRI equipment, optimized for cardiac imaging, has achieved great accuracy [23,24]. Acquisition sequences specifically developed to image the heart movement, Steady State Free Precession (SSFP), can demonstrate a reciprocal sliding motion between the neoplasm and nearby cardiovascular structures, which, whenever present, represents a suggestive sign of absent infiltration [25]. Moreover, the cine-MRI signal on SSFP sequences is nulled by a fat–water interface when included in the same voxel, resulting in a black line signal void at the contact interface. It allows us to exploit a technical artifact, the so-called “India Ink” artifact, as a clinically useful tool to highlight the presence of preserved perivascular or epicardial fat (adipose tissue cleavage) as a sign of non-infiltration of the great vessels or the pericardium [26,27]. 

Given such a background, we sought to compare the diagnostic accuracy of contrast-enhanced CT scans and cine-MRI in detecting mediastinal masses’ infiltration of nearby cardiovascular structures.

## 2. Materials and Methods

### 2.1. Study Design and Population

We retrospectively retrieved from the Humanitas Research Hospital (Rozzano, Italy) electronic archive the clinical and surgical records of 50 consecutive patients (21 females, 42%) who underwent surgical excision at our Institution from 1 January 2009 to 31 December 2019. 

We included patients with a performance status ECOG 0–2 affected by a malignant tumor of the lung or the mediastinum involving the mediastinal cardiovascular structures and who were candidates for surgery. We excluded patients with unresectable disease or with distant metastasis.

All patients preoperatively underwent contrast-enhanced total body CT scan for tumor staging and characterization, followed by cine-MRI to integrate CT findings whenever clinicians were not confident about the radiological CT findings on the presence/absence of mediastinal structures infiltration.

We compared CT and cine-MRI report findings regarding infiltration with the surgical and pathological post-surgical description, which was used as a reference standard for invasion detection. A multidisciplinary discussion was performed in all cases including radiologists, thoracic surgeons and cardiovascular surgeons. We also collected detailed information about the surgical intervention specifying the mediastinal cardiovascular structure resected and the surgical strategy adopted. We also evaluated how imaging results affected surgical decisions in terms of changes in the surgical plan to achieve radicality without harming patients.

Imaging protocol and analysis

All patients underwent a non-ECG-gated contrast-enhanced chest CT with a >/=64-slice detector. Scans were mainly performed at our Institution on a 256-row CT scanner, but were not repeated in case of a recent good-quality scan from a referral hospital. Images were acquired before and after iodinated contrast medium administration with a slice thickness of 0.625 mm, and multi-planar reconstructions were performed. They also underwent a dedicated cardiac cine-MRI protocol only based on ECG-gated SSFP sequences acquired in the axial, coronal and sagittal planes to cover the entire neoplastic mass (TE 1.3; TR 30.8; flip angle 59°). Images were acquired with a 7 mm slice thickness and no gap. We reconstructed 30 phases per heartbeat with a scan duration of about 7–10 s for every slice; the number of slices per apnea was adapted according to patient-specific ability to hold their breath. No contrast was administered to shorten the protocol. Images were reviewed at the end of the orthogonal acquisitions, and supplemental SSFP in non-orthogonal planes were eventually prescribed (perpendicular to the interface between mass and vessel/myocardium) to facilitate the differential diagnosis between partial volume and real infiltration. 

Contrast-enhanced chest CT images were evaluated by a senior (>20 years of experience) general radiologist, while cine-MRI was evaluated by another experienced (>15 years of experience) cardio-thoracic radiologist.

An additional analysis was conducted to compute inter- and intra-observer agreement over a homogeneous subset population of 14 random patients included in the study. The same experienced general and cardio-thoracic radiologists who signed the reports reviewed CT images and cine-MRI, respectively, to estimate intra-observer agreement. At the same time, another two general and cardio-thoracic radiologists reviewed the same images to assess inter-observer agreement.

Infiltration was evaluated on the basis of the radiological reports of both CT and cine-MRI. On the contrast-enhanced CT scans, infiltration was reported when adipose tissue cleavage between the mass and the surrounding cardiovascular structures was absent. On cine-MRI, direct invasion was confirmed by the absence of the sliding motion between the mass and the heart and—for extracardiac sites—by the absence of the “India-Ink” artifact between the mass and the adjacent structures (Figure 1 and Figure 2). The surgical approach was defined after a multidisciplinary meeting where CT and MRI images of a single case were studied by radiologists and surgeons. 

### 2.2. Statistical Analysis

Statistical analysis was performed in Python (version 3.9) with Scipy Library (version 1.9.3). All continuous variables were tested for normality distribution with the Shapiro–Wilk test. Normally distributed features were compared through a two-sided *t*-test, whereas non-normally distributed features were tested with the two-sided Mann Whitney U test. Discrete features were compared with the χ^2^ test. Statistical significance was set as a *p*-value < 0.05. The diagnostic performance of imaging techniques was assessed by computing sensitivity (Sn), specificity (Sp), accuracy, negative predictive value (NPV) and positive predictive value (PPV). The assessment of the risk of infiltration and changes in the surgical plan was calculated by computing the odds ratio (OR) from the binary logistic regression model, with a 95% confidence interval (CI).

Inter- and intra-observer agreement was calculated using the Cohen kappa score.

## 3. Results

The final study population included 50 patients (22 females) with a mean age of 59 ± 14 years (range: 19–81). The diagnoses at histology were as follows: lung cancer in 23 patients (46%), thymic neoplasms in 11 (22%), sarcoma in 8 (16%) and other neoplasms in 8 (16%) (Table 1).

Cine-MRI and CT scans demonstrated mediastinal cardiovascular structures infiltration in thirty-five (70%) and nine patients (30%), respectively. The pathology report confirmed the infiltration in 21 (42%) out of 50 patients (Table 2).

Cine-MRI showed a higher sensitivity compared to CT scans in detecting cardiovascular invasion (91% vs. 16%, *p* < 0.001), as well as greater accuracy (66% vs. 50%, *p* < 0.001) and negative predictive value (93% vs. 54%, *p* < 0.001), allowing for a better preoperative staging of mediastinal masses and aiding better preoperative surgical planning (Figure 3).

Cine-MRI specificity, on the contrary, turned out to be lower than CT scan specificity (46% vs. 79%, *p* < 0.001) due to the higher number of false-positive cases. According to the imaging study, the surgical approach was modified in 23 (46%) patients: in 15 (30%) cases, the intervention was performed with a cardiac surgeon because of the involvement of the cardiac chambers, ascending aorta or pulmonary trunk. Eight (16%) cases required a vascular surgeon for the invasion of the superior vena cava or descending aorta (Table 1). 

In all these cases, the interventions were performed in the cardiac surgery department with a dedicated team of anesthesiologists, surgeons and nurses. This setting provided the possibility to perform the intervention with extracorporeal circulation with a normothermic cardiopulmonary bypass. In three cases, a right pneumonectomy with carinal resection was performed. To provide cardiac and respiratory support, a heparin-coated percutaneous cannula (16 Fr) was placed in the right jugular vein under ultrasound guidance, and a 24 Fr cannula was placed with a percutaneous technique in the right femoral vein cannula for the veno-venous-ECMO circuit. 

When the superior vena cava was involved, a venous-venous shunt was used to allow resection, while reconstruction was undertaken with a cryopreserved cadaveric aorta. No statistically significant result was found looking at the odds ratio (OR) for the propensity to change the surgical plan for both modalities, even though cine-MRI also demonstrated a greater influence on the surgeon’s decision, with an OR of 1.32 (*p*-value = 0.074) vs. the CT scan OR of 1.22 (*p*-value = 0.277).

When the CT scan and cine-MRI were discordant and cine-MRI was negative for invasion, histological results were 100% concordant with the latter (Figure 4).

When the CT scan and cine-MRI were discordant and the CT scan was negative for invasion, a more balanced distribution was seen in the pathological records (59% invasion and 41% no invasion).

When the CT scan and cine-MRI were concordant about no infiltration, in 92%, this finding was confirmed at histology.

When the CT scan and cine-MRI were concordant about infiltration (*n* = 5) and surgery was performed, 60% of patients had infiltration, whereas, in two (40%) cases, (suspect involvement of the right inferior pulmonary vein in one case and right atrium in the other) infiltration was absent.

Intra-observer agreement showed a Cohen kappa score of 0.86 for cine-MRI and 0.32 for CT scans. Inter-observer agreement showed a Cohen kappa score of 0.72 for cine-MRI and 0.19 for CT scans. 

## 4. Discussion

The ability to diagnose the infiltration of the heart/great vessels district is critical to assess resectability and to shape the best surgical approach, with a considerable impact on the patient’s management. The choice to plan an extracorporeal circulation or a superior vena cava replacement should be made before surgery to prepare dedicated surgeons (i.e., vascular or cardiac surgeons) and a properly equipped surgical theatre [10,11].

In our study, we aimed to re-shape the pre-operative planning of patients with uncertain or equivocal CT findings using SSFP sequences in cine-MRI, which currently represent the gold standard for the volume, function and mass assessment of both cardiac ventricles [28]; moreover, SSFP sequences provide good spatial resolution and optimal tissue contrast.

The sliding motion of the mass with respect to pulsating cardiovascular structures was used to discriminate the presence/absence of the infiltration of the cardiac chambers, while the “India Ink” artifact discriminated the presence/absence of the infiltration of the great vessels. In this specific setting, the “India Ink” artifact, by means of the magnification of the interface between the mass and the adventitial fat layer, overcomes the intrinsic limits of standard MRI in depicting structures with a thickness far below the spatial resolution of the modality [29].

Already, in 2005, Seo JC et al. reported that cine-MRI could provide additional information compared to contrast-enhanced CT scans, improving the accuracy of preoperative staging for predicting the cardiovascular invasion of a thoracic mass in the mediastinum [25,30]. Similarly, the main finding of our study is that cine-MRI provides significantly higher diagnostic accuracy in the identification of infiltration compared to contrast-enhanced CT scans when evaluating the invasion of the cardiac/great vessel district. From a surgical perspective, the significantly better negative predictive value of cine-MRI is of utmost importance for thoracic surgeons, allowing greater confidence in the final decision on whether and how to approach the surgical procedure (Figure 5). 

Moreover, we further demonstrated that the propensity to change surgical plans, even without statistically significant results, tended to be greater when based on cine-MRI than with contrast-enhanced CT scans. In 15 of the 20 true-positive patients showing infiltration via cine-MRI, surgeons decided to tailor the surgical approach. When MRI excluded the infiltration of cardiovascular structures, the use of ECC or prosthetic material was always avoided.

In our study, CT scans have a very low sensitivity in detecting cardiovascular invasion (14%), far below the current literature data. This finding may be related to the intrinsic selection bias of the study population that was enrolled on the basis of equivocal or uncertain findings at the CT scan stage. 

Our results also outline the low specificity of cine-MRI in detecting infiltration (41%), giving a consistent number of false-positive cases. The adopted MR study protocol was pre-specified in order to simplify the image acquisition and to generate more reproducible results, with acquisition in axial, coronal and sagittal planes covering the mediastinal tumor. Therefore, not all the cleavage regions were addressed with specifically oriented planes. The orientation of the scanning plane perpendicular to the cleavage region of interest is crucial for the “India Ink” hypointense rim formation; tortuous interfaces can therefore limit the “India Ink” artifact formation because of partial volume effects, thus increasing the number of false-positive cases.

We also demonstrated that, over a homogeneous sample of fourteen patients (seven with and seven without infiltration at histology), cine-MRI shows higher inter- and intra-observer agreement, further reinforcing its clinical power from the surgical point of view. This finding can be explained by the more robust diagnostic evidence of infiltration: while in contrast-enhanced CT scans, the infiltration parameter was simply the presence/absence of the adipose tissue cleavage between the mass and the mediastinal cardiovascular structures and the percentage of the circumference with adhesion in the case of aortic invasion, cine-MRI infiltration criteria are more precise, even simple, and are also already tested and approved in the current literature [31]. The “India Ink” artifact, as clearly demonstrated, magnifying the presence in the same voxel of both fat-rich and water-rich elements, highlighted interfaces otherwise below the spatial resolution of the MRI scanners [32].

Although cine-MRI showed higher inter- and intra-observer agreement than CT scans, we still noted variability in the interpretation of imaging results: cine-MRI intra-observer vs. inter-observer agreement = 086 vs. 0.72. This result can be explained by the different seniority of the readers involved when assessing inter-observer agreement. The reader’s seniority and specific expertise is a known factor affecting the accuracy of cardiac MRI analysis [33], which could partially affect the consistency of the findings.

### Study Limitations

Our study presents some limitations. 

We collected a limited number of resectable cases characterized by uncertain findings about cardiovascular infiltration via CT scans, but with a heterogeneous distribution of tumor histology.

The limited sample size hampered the possibility of performing district- or pathology-specific sub-analysis. Moreover, results are limited to patients with uncertain findings from CT scans, and they cannot be extended to the entire population of patients with mediastinal masses.

In an additional questionable method of this study, we did not perform ECG-gated contrast-enhanced chest CT scans, possibly leading to underestimation of the latter’s accuracy in infiltration detection. ECG gating has been demonstrated to increase the spatial resolution of cardiac and vascular structures, but not of tissue contrast, which is crucial for the diagnosis of infiltration. We tested, in three cases, both ECG-gated CT scans and cine-MRI as second-line exams; there was no diagnostic improvement with gated CT scans due to the intrinsically low tissue contrast of the method. Therefore, we used only cine-MRI as a second-line examination. 

The ECG trigger, despite possibly improving CT scans’ discriminative power over cardiac chambers [32], does not improve contrast resolution, which is what the “India Ink” artifact magnifies in cine-MRI, especially when evaluating vessel infiltration. Performing additional ECG-triggered CT scans, after a previous contrast-enhanced thoracic CT scan, implies a non-negligible adjunctive radiation dose delivered to the patient, which, given the high quality of non-triggered images and the minimal improvement in contrast resolution, would not be justified. Moreover, due to the retrospective design of the study, we decided to conduct a real-life examination, evaluating contrast-enhanced CT-scan accuracy as the gold standard for the T staging of masses involving the mediastinum. 

Further studies adopting new technologies such as spectral CT or photon-counting detectors may change the possible role of ECG-gated CT in the future. Further research with more homogeneously studied patient populations and newer imaging techniques is warranted to validate the findings and improve the diagnostic accuracy of cine-MRI in assessing mediastinal masses.

## 5. Conclusions

Breath-hold, ECG-gated cine-MRI using SSFP sequences is feasible in all patients with advanced primary mediastinal masses or lung cancers with suspected cardiovascular involvement, and it outperforms conventional contrast-enhanced chest CT scans in the preoperative assessment by means of better sensitivity, accuracy and—above all—negative predictive value. For this reason, it should be considered in the preoperative staging and evaluation of patients with equivocal or uncertain findings at the pre-operative CT scan stage to better identify infiltration and correctly plan surgical excision.

## Figures and Tables

**Figure 1 diagnostics-14-01682-f001:**
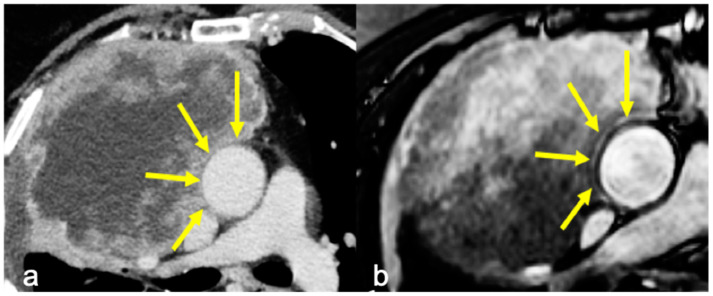
(**a**) Axial contrast-enhanced CT showing a fat cleavage between the ascending aorta and the tumor and (**b**) cine-MRI showing “India Ink” artifact between the two (yellow arrows). Infiltration of the aorta was absent at surgery.

**Figure 2 diagnostics-14-01682-f002:**
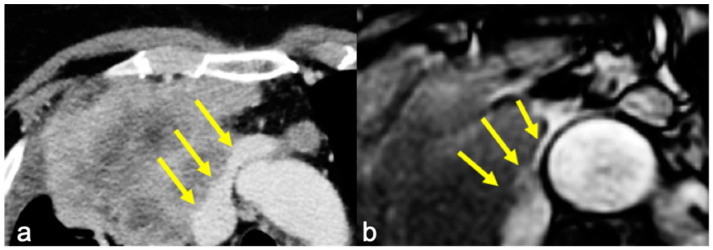
(**a**) Axial contrast-enhanced CT showing irregular margins at the left innominate vein and superior vena cava confluence and (**b**) cine-MRI of the same patient with no “India Ink” artifact between the mass and the vessels, suspicious for infiltration (yellow arrows). Infiltration of the left brachiocephalic vein was confirmed at surgery.

**Figure 3 diagnostics-14-01682-f003:**
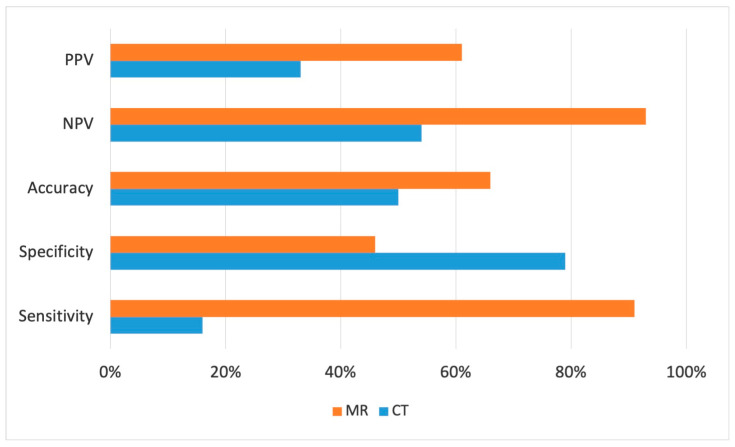
Histogram representation of CT (blue) and cine-MRI (orange) comparison of sensitivity, specificity, accuracy, positive and negative predictive values in detecting infiltration.

**Figure 4 diagnostics-14-01682-f004:**
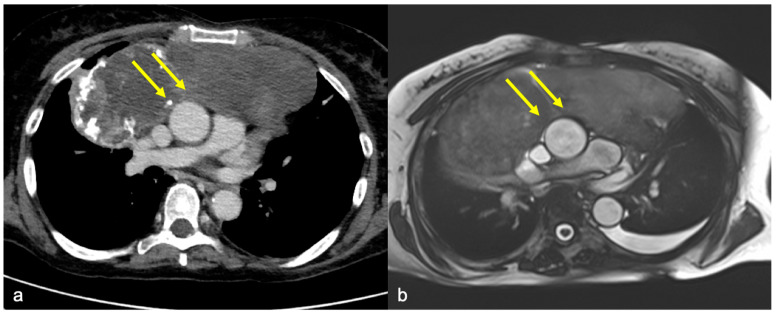
The case of a 50 y.o. patient affected by a mediastinal sarcoma in which pre-operative chest CT posed the suspicion of ascending aorta infiltration by the mass: image (**a**) shows the absence of adipose tissue cleavage between the mass and the aorta (yellow arrows) in the axial plane. Cine-MRI (image (**b**)) shows the “India Ink” artifact between the vessel and the mass (yellow arrows), excluding the presence of infiltration. Intraoperative evidence and final histological report confirmed cine-MRI findings, excluding vascular infiltration.

**Figure 5 diagnostics-14-01682-f005:**
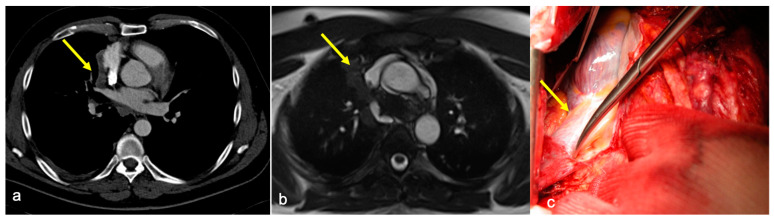
The case of a 49 y.o. patient affected by a thymoma in which pre-operative chest CT posed the suspicion of superior vena cava infiltration by the mass: image (**a**) shows the absence of adipose tissue cleavage between the mass and the vessel (yellow arrow) in the axial plane. Cine-MRI (image (**b**)) excluded the infiltration according to the presence of the “India Ink” artifact between thymoma and superior vena cava (yellow arrow). Image (**c**) shows an intra-operatory picture of the suspicious site of infiltration, the superior vena cava (yellow arrow), which was finally excluded by pathological report. In the adjunctive material, pure axial SSFP demonstrated reciprocal sliding motion and the “India Ink” artifact between the mass and the superior vena cava.

**Table 1 diagnostics-14-01682-t001:** Patient characteristics.

	N°	%
Population	50	100
Female	21	42
Mean age	59 ±14	
Histology		
NSCLC	23	46
Thymic tumor	11	22
Sarcoma	8	16
Other neoplasms	8	16
Mediastinal imaging involvement (CT)		
Right atrium	8	16
Left atrium	9	18
Pulmonary	7	14
Superior vena	7	14
Left ventricle	5	10
Ascending aorta	3	6
Descending	4	8
Pericardium	7	14
Combination surgery	23	46
Cardiac surgeon	15	30
Vascular surgeon	8	16

**Table 2 diagnostics-14-01682-t002:** Schematic representation of confusion matrix for both CT and cine-MRI with respect to the pathological post-surgical diagnosis.

*n* = 50	Infiltration	No-Infiltration
CT	9	41
Cine-MRI	35	15
Pathological report	21	29

## Data Availability

The data presented in this study are available on request from the corresponding authors. The data are not publicly available due to privacy and ethics.

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
