# Peer review of "Role of Cine-Magnetic Resonance Imaging in the Assessment of Mediastinal Masses with Uncertain/Equivocal Findings from Pre-Operative Computed Tomography Scanning"

_diagnostics, 2024, doi:10.3390/diagnostics14151682_

Round 1
Reviewer 1 Report
Comments and Suggestions for Authors
The article presents a few limitations, as outlined in the study itself:
1. The study's retrospective nature may introduce selection bias, as the cases were selected based on uncertain or equivocal CT findings, which might not represent the general population of patients with mediastinal masses.
2. The study included a limited number of cases (50 patients) with a heterogeneous distribution of tumor histology. This small and varied sample size limits the ability to perform detailed sub-analyses specific to different tumor types or anatomical regions.
3. The study found that cine-MRI had a low specificity (41%) in detecting infiltration, resulting in many false-positive cases. This limitation is partly attributed to the need for the correct orientation of the scanning plane perpendicularly to the cleavage region of interest, which was not always achieved.
4. The study did not use ECG-gated contrast-enhanced chest CT, which could have improved the temporal resolution of the CT images. This omission might have led to underestimating CT's accuracy in detecting infiltration.
5. Performing additional ECG-triggered CT scans after a previous contrast-enhanced thoracic CT would have resulted in a non-negligible additional radiation dose to the patients, which the study aimed to avoid.
6. Although cine-MRI showed higher inter- and intra-observer agreement than CT, the study still noted variability in the interpretation of imaging results, which could affect the consistency of the findings.
7. The study mentions using the "India Ink" artifact in cine-MRI, which relies on the presence of fat-rich and water-rich elements in the same voxel. This artifact can be influenced by partial volume effects, especially in tortuous structures, potentially leading to diagnostic inaccuracies.
8. The study aimed to reflect real-life clinical practice by not repeating recent good-quality CT scans from referral hospitals, which might have introduced variability in the imaging quality and protocols.
These limitations highlight the need for further research with larger, more homogeneous patient populations and advanced imaging techniques to validate the findings and improve the diagnostic accuracy of cine-MRI in assessing mediastinal masses.
Comments on the Quality of English Languageminor
Reviewer 2 Report
Comments and Suggestions for Authors
The introduction provides a comprehensive background on the diagnostic challenges posed by malignant neoplasms in the mediastinum, mainly when they are in contact with cardiovascular structures. The authors effectively justify the need for alternative imaging modalities like cine-MRI by highlighting the limitations of contrast-enhanced CT scans in some instances. The introduction is well-referenced and includes relevant literature. However, it could benefit from a more detailed discussion of the specific advancements in cine-MRI technology that make it a promising tool.
The research design is appropriate for the study's aims. The retrospective analysis of 50 patients provides a solid sample size for evaluating the diagnostic performance of cine-MRI compared to CT. Including intraoperative findings and histological post-surgical reports as reference standards strengthens the validity of the results. However, the study could improve by providing more details on the criteria used to select patients and any potential biases this might introduce (10.1148/rg.230091, 10.1109/ISBI53787.2023.10230686).
The methods section is adequately described, detailing the imaging protocols for both CT and cine-MRI. The explanation of how infiltration was assessed using both modalities is clear. However, the paper would benefit from more information on the statistical methods used to compare diagnostic performance (10.1007/978-3-031-28524-0_7, 10.26044/ecr2023/C-16014), particularly how p-values were calculated and any adjustments made for multiple comparisons. Additionally, detailed patient demographics and clinical characteristics would provide more context for the findings.
The results are presented, with appropriate use of tables and figures to illustrate key points. The significant improvement in sensitivity, accuracy, and negative predictive value of cine-MRI over CT is well-documented. The confusion matrix and the inter and intra-observer agreement data are precious in demonstrating the reliability of cine-MRI. However, the discussion around the lower specificity of cine-MRI could be expanded to explore potential reasons for false positives and how this might be addressed in future studies or mitigated by technology (10.1186/s40644-023-00560-z, 10.1109/ISBI53787.2023.10230448).
The results support the conclusions and highlight the clinical implications of using cine-MRI for preoperative assessment of mediastinal masses. The paper effectively argues for the inclusion of cine-MRI in the diagnostic pathway for patients with equivocal CT findings, emphasizing its higher sensitivity and negative predictive value. The conclusion could be strengthened by discussing the potential limitations of the study in more detail and suggesting areas for future research.
Comments on the Quality of English LanguageThe manuscript is generally well-written, but it requires minor editing for the English language to enhance clarity and readability. There are a few grammatical errors and awkward phrasings that need to be addressed. Improving these aspects will ensure that the content is more comprehensible and professional. Overall, the quality of English is good, but careful proofreading and minor revisions will be beneficial.
Reviewer 3 Report
Comments and Suggestions for Authors
The manuscript targets to evaluate the diagnostic accuracy of cine-MRI in detecting cardiovascular infiltration by mediastinal tumors, compared to contrast-enhanced CT, in cases with uncertain findings.
Title:
Spell out all acronyms the first time that they are used (e.g.: MRI, CT)
Material & Methods
Provide more demographic details about the patient population, such as age range, comorbidities, and types of neoplasms.
The protocol for the contrast-enhanced CT is not fully detailed.
I recommend that the inclusion and exclusion criteria be detailed more comprehensively.
A clear definition of the inclusion and exclusion criteria for patient selection is missing.
Conclusion:
Mention the need for further research to confirm these findings across larger, more diverse populations and to explore long-term outcomes related to the use of cine-MRI in preoperative staging.
Round 2
Reviewer 1 Report
Comments and Suggestions for Authors
Thank you for the edits.
no more revisions
Comments on the Quality of English Languageminor
Reviewer 3 Report
Comments and Suggestions for Authors
The authors responses are accepted.